# GreedyCAS: Unsupervised Scientific Abstract Segmentation with Normalized Mutual Information

**Yingqiang Gao**[†], **Jessica Lam**[†], **Nianlong Gu**[‡]
**Richard H.R. Hahnloser**[†]

[†]Institute of Neuroinformatics, University of Zurich and ETH Zurich, Switzerland
`{yingqiang.gao, lamjessica, rich}@ini.ethz.ch`
[‡]Linguistic Research Infrastructure, University of Zurich, Switzerland
`nianlong.gu@uzh.ch`

## Abstract

The abstracts of scientific papers typically contain both premises (e.g., background and observations) and conclusions. Although conclusion sentences are highlighted in structured abstracts, in non-structured abstracts the concluding information is not explicitly marked, which makes the automatic segmentation of conclusions from scientific abstracts a challenging task. In this work, we explore Normalized Mutual Information (NMI) as a means for abstract segmentation. We consider each abstract as a recurrent cycle of sentences and place two segmentation boundaries by greedily optimizing the NMI score between the two segments, assuming that conclusions are strongly semantically linked with preceding premises. On non-structured abstracts, our proposed unsupervised approach GreedyCAS achieves the best performance across all evaluation metrics; on structured abstracts, GreedyCAS outperforms all baseline methods measured by $P_k$. The strong correlation of NMI to our evaluation metrics reveals the effectiveness of NMI for abstract segmentation.[1]

## 1 Introduction

Abstracts of scientific papers are short texts that summarize the findings reported in the body text (Bahadoran et al., 2020). A well-formulated abstract forms a scientific inference that extends from premises (e.g., shared knowledge, experimental evidence, or observation) to conclusions (e.g., suggestions, claims, Ripple et al. (2012)). A splitting of an abstract into a conclusion segment and a premise segment can help readers better comprehend how conclusions are drawn (Bahadoran et al., 2020) and is of interest for downstream research tasks such as argument generation (Schiller et al., 2021), knowledge retrieval (Hua et al., 2019), opinion analysis (Hulpus et al., 2019), and text summarization (Cho et al., 2022).

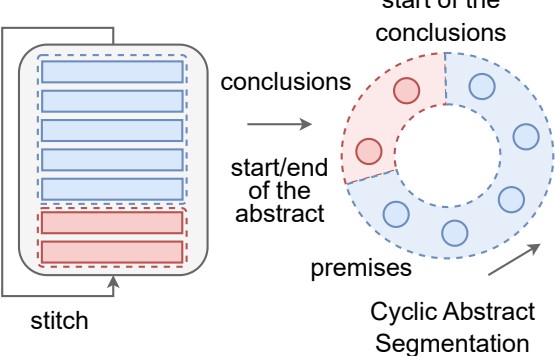

Figure 1: **Left**: An abstract that contains premise (blue) and conclusion (red) sentences. The task is to identify the start and end segmentation boundaries of the conclusions. **Right**: We regard each abstract as a recurrent cycle of sentences by stitching its start and end together. Best view in color printing.

Many abstracts, especially those from the biomedical domain, are structured to help the reader extract the conclusions (e.g., abstracts follow the IMRaD format (Nair and Nair, 2014; Dernoncourt and Lee, 2017) or the CONSORT format (Hopewell et al., 2008)). In contrast, abstracts from many other research domains do not explicitly indicate the position of conclusions, which means that readers must perform the potentially cognitively demanding task of identifying the conclusions themselves. We are therefore interested in probing approaches for splitting scientific abstracts into conclusion and premise sentences.

Existing text segmentation approaches (Somasundaran et al., 2020; Lo et al., 2021; Barrow et al., 2020; Koshorek et al., 2018) can be applied to scientific abstract segmentation. However, fine-tuning such models typically requires large amounts of labelled data that are expensive to collect. In contrast, unsupervised approaches require no annotated data and can segment large numbers of texts with minimal human involvement. Thus, we primarily test

---

[1]Code and data available at `https://github.com/CharizardAcademy/GreedyCAS.git`

unsupervised frameworks for segmenting scientific abstracts.

Given a set of abstracts, we want to determine their splits into premise segments and conclusion segments. Combining the premises from all abstracts gives us a premise set, and similarly, we obtain a conclusion set from combining all conclusion segments. We hypothesize that the abstracts are best segmented when the **N**ormalized **M**utual **I**nformation (NMI) between the conclusion set and premise set is maximized. Our intuition is that conclusions follow from the remainder of the abstract, which is a redundancy that is well captured by mutual information. To maximize NMI, we use an exhaustive greedy approach that iterates over all abstracts and determines the best segmentation for each. To test how NMI deals with a known text boundary, the end of an abstract, we stitch the start and end of each abstract together to form a cycle, then select two segmentation boundaries with constraints based on prior knowledge (see Figure 1). We name our approach **G**reedy **C**yclic **A**bstract **S**egmentation (GreedyCAS).

To test our proposed approach, we create two datasets. One dataset comprises non-structured abstracts with human-annotated conclusion sentences. The other dataset contains structured abstracts in which conclusion sentences have been explicitly marked by the authors of the abstract.

Our main **contributions** are as follows:

- We propose GreedyCAS, an unsupervised approach for scientific abstract segmentation that optimizes NMI.

- On a dataset of non-structured abstracts, we show that GreedyCAS achieves promising segmentation results.

- We find a strong correlation between NMI and other evaluation metrics, in support of NMI being useful for segmentation.

## 2 Related Works

Abstract segmentation is a particular case of text segmentation. The task of text segmentation is to insert separation markers into the text such that the segmented fragments are topically coherent and comprehensive (Hazem et al., 2020). Traditional methods can be categorized into supervised and unsupervised methods.

Unsupervised approaches (Alemi and Ginsparg, 2015) usually make use of metrics based on textual coherence or topic contiguity and take the following strategies: 1) to greedily seek the best segmentation based on text similarity at each step (Choi, 2000; Hearst, 1994); and 2) to iteratively approach a global optimum of a segmentation objective (e.g. semantic relatedness) via dynamic programming (Fragkou et al., 2004; Bayomi and Lawless, 2018). These methods use similarity measures or lexical frequencies between segments to determine segmentation boundaries or convert the inter-sentence similarities into a semantic graph and perform graph search (Glavaš et al., 2016) to find the optimal segmentation.

Supervised methods are usually deployed when sufficiently many annotated examples are available. These methods typically use language models to encode sentences and perform binary classification to predict whether a sentence is on the segmentation boundary (Somasundaran et al., 2020; Banerjee et al., 2020; Aumiller et al., 2021; Badjatiya et al., 2018; Lukasik et al., 2020; Koshorek et al., 2018). Banerjee et al. (2020) fine-tuned a hierarchical sentence encoder using structured abstracts to classify sentences into discourse categories (BACKGROUND, TECHNIQUE, and OBSERVATION). In general, supervised approaches achieve good performance, but they consume a large amount of annotated data, which is expensive to collect. In this work, we set out to test generic unsupervised approaches that require no training data.

## 3 Methodology

We formulate the task of segmenting scientific abstracts as follows. Given an abstract $A = (s_i)_{i=1}^n$ containing $n$ sentences, we define $G^A = \{g_j^A\}_{j=1}^{m_A}$ as the set of all $m_A$ possible segmentations of $A$. Each segmentation $g_j^A = (P_j^A, C_j^A)$ contains a premise segment $P_j^A$ and a conclusion segment $C_j^A$ with boundaries given by two indices $\alpha_j^A, \xi_j^A \in \mathbb{N}_{1:n}$ (range of integers from 1 to $n$):

$$C_j^A = \{s_i \in A \mid \alpha_j^A \leq i \leq \xi_j^A\},$$
$$P_j^A = \{s_i \in A \mid s_i \notin C_j^A\}.$$

Associated with a corpus $\mathbb{A} = \{A_i\}_{i \in \mathbb{N}_{1:k}}$ of $k$ abstracts, there is a set of $\mathbb{G} = \{G^{A_i}\}_{i \in \mathbb{N}_{1:k}}$ of $\prod_{i=1}^k m_{A_i}$ possible segmentations. Searching for the best ensemble of segmentations of $\mathbb{A}$

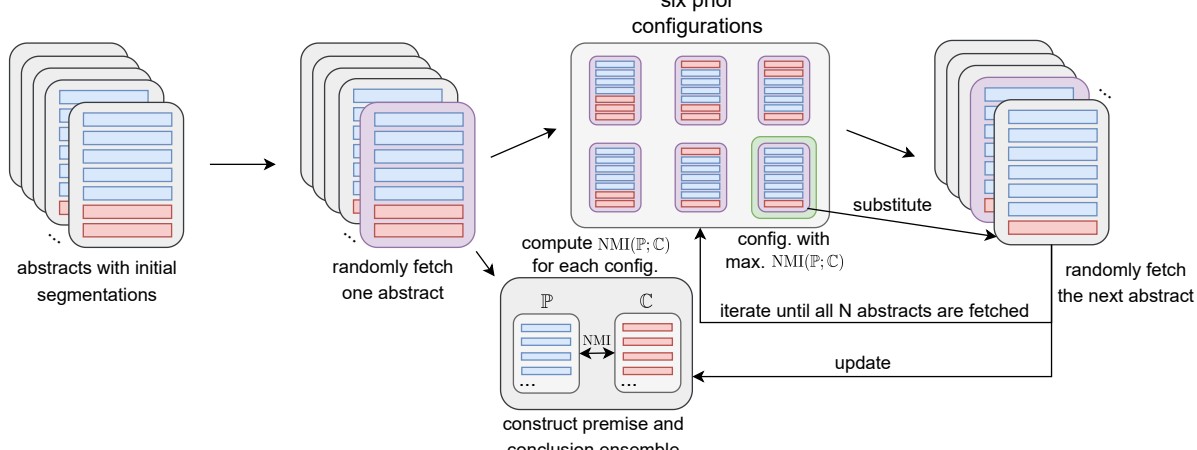

six prior configurations

abstracts with initial segmentations → randomly fetch one abstract → compute $\mathrm{NMI}(\mathbb{P};\mathbb{C})$ for each config. → config. with max. $\mathrm{NMI}(\mathbb{P};\mathbb{C})$ → substitute → randomly fetch the next abstract

iterate until all N abstracts are fetched

update

construct premise and conclusion ensemble

Figure 2: Our proposed GreedyCAS pipeline for segmenting scientific abstracts. Premise sentences are colored in blue, whereas conclusion sentences are colored in red. Best view in color printing.

within $\mathbb{G}$ may involve exhaustive enumeration over $\prod_{i=1}^{k} m_{A_i}$ segmentations, which is impossible under limited computational costs. Therefore, in this work, we concentrate on greedily approaching a reasonably good segmentation that is a tight lower bound of the actual global optimum.

## 3.1 Cyclic Abstract Segmentation

To reduce the search space, we make two assumptions: firstly, that conclusion sentences are located at the end of abstracts; and secondly, that each abstract contains at most three conclusion sentences. This results in $m = 6$ possible segmentations per abstract. The segmentations of an example abstract with $n = 7$ sentences are depicted in Table 1.

Because scientific abstracts typically end with conclusion sentences, we expect the stitching point of our cyclic abstracts to form a boundary. In other words, we read out the segmentation of interest from the first segment boundary $\alpha_j^A$; the second segment boundary $\xi_j^A$ we expect to coincide with the abstract end. Thus, by optimizing the second segment boundary $\xi_j^A$, we perform a sanity check that the unsupervised segmentation method is capable of detecting the abstract end, which is a natural boundary of the abstract.

## 3.2 Normalized Mutual Information

Our next step is to choose an optimization objective for the greedy search. Inspired by work on text summarization (Padmakumar and He, 2021) and birdsong analysis (Sainburg et al., 2019), we explore mutual information as the optimization objective.

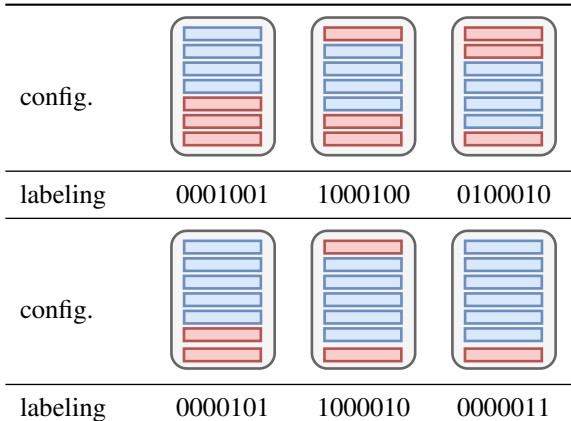

| | | | |
|---|---|---|---|
| config. | | | |
| labeling | 0001001 | 1000100 | 0100010 |
| config. | | | |
| labeling | 0000101 | 1000010 | 0000011 |

Table 1: Labeling of segmentation boundaries for an example abstract with seven sentences, among which at most three sentences are conclusion sentences (colored in red). 1 indicates a sentence on the segment boundary, whereas 0 indicates a non-boundary sentence. We aim to reduce the complexity by searching within the six possible segmentations for each abstract.

Mutual information $I(X;Y)$ is a measure of the absolute reduction in information uncertainty (in bits) for a random variable $X$ after observing another correlated random variable $Y$. Our proposed greedy approach is based on the assumption that the uncertainty of the conclusion is minimized after the premise is observed, i.e. that the segmentation maximizes mutual information.

We denote $\mathbb{C} = \{C_{j_i}^{A_i}\}_{i \in \mathbb{N}_{1:k}}$ as a possible *conclusion ensemble* spanned by all conclusion segments from the $k$ abstracts, and $\mathbb{P} = \{P_{j_i}^{A_i}\}_{i \in \mathbb{N}_{1:k}}$ as one possible *premise ensemble* obtained in the same way. Note that the segmentation $j_i$ can be different for each abstract $A_i$.

The task can now be formulated as follows: given a corpus of abstracts $\mathbb{A}$, determine the premise $\mathbb{P}$ and conclusion $\mathbb{C}$ ensembles that maximize the mutual information $I(\mathbb{P}; \mathbb{C})$.

More formally, we compute $I(\mathbb{P}; \mathbb{C})$ as follows:

$$I(\mathbb{P}; \mathbb{C}) =$$
$$\sum_{A_i \in \mathbb{A}} \sum_{w_p \in P_{j_i}^{A_i}} \sum_{w_c \in C_{j_i}^{A_i}} p(w_p; w_c) \log \frac{p(w_p; w_c)}{p(w_p)p(w_c)},$$

where $w_p$ and $w_c$ are unigram tokens in the $i$-th premise segment $P_{j_i}^{A_i}$ and the $i$-th conclusion segment $C_{j_i}^{A_i}$, respectively. $p(w_p; w_c)$ indicates the joint probability of the premise word $w_p$ appearing in the premise segment $P_{j_i}^{A_i}$ and the conclusion word $w_c$ appearing in the conclusion segment $C_{j_i}^{A_i}$. $p(w_p)$ and $p(w_c)$ denote marginal probabilities. Making use of language modeling statistics, we compute the marginal probabilities as follows:

$$p(w_p) = \frac{c(w_p, \mathbb{P})}{\sum_{w_p'} c(w_p', \mathbb{P})}$$
$$p(w_c) = \frac{c(w_c, \mathbb{C})}{\sum_{w_c'} c(w_c', \mathbb{C})},$$

where $c(w, \mathbb{P})$ denotes the number of occurrences of $w$ within the tokenized premise segments in $\mathbb{P}$ and $w_p'$ is a token from the premise segment of any abstract. The terms $c(w, \mathbb{C})$ and $w_c'$ are defined analogously.

The joint probability is then computed as

$$p(w_p; w_c) = \frac{\sum_{i=1}^{k} c(w_p, P_{j_i}^{A_i}) c(w_c, C_{j_i}^{A_i})}{\sum_{(w_p', w_c')} c(w_p', \mathbb{P}) c(w_c', \mathbb{C})},$$

Because mutual information is an unbounded measure that increases with the size of $\mathbb{A}$, it is not directly comparable across different $\mathbb{P}$ and $\mathbb{C}$ ensembles (Poole et al., 2019). We therefore normalize $I(\mathbb{P}; \mathbb{C})$ by mapping it onto the interval $[0, 1]$ and use Normalized Mutual Information (NMI) as the final optimization objective.

Taken from Kvålseth (2017), we compute NMI as follows:

$$\mathrm{NMI}(\mathbb{P}; \mathbb{C}) = \frac{I(\mathbb{P}; \mathbb{C})}{\mathcal{U}_a}$$

where $\mathcal{U}_a$ denotes the non-decreasing theoretical upper bound of $I(\mathbb{P}; \mathbb{C})$ and is parametrized by the $a$-order arithmetic mean

$$\mathcal{U}_a = \left( \frac{\mathcal{U}_\mathbb{P} + \mathcal{U}_\mathbb{C}}{2} \right)^{1/a}.$$

Here, we have

$$\mathcal{U}_\mathbb{P} = -\sum_{w_p} p(w_p) \log p(w_p) = H(\mathbb{P})$$

and

$$\mathcal{U}_\mathbb{C} = -\sum_{w_c} p(w_c) \log p(w_c) = H(\mathbb{C})$$

essentially being the entropy of the premise ensemble and the conclusion ensemble, respectively. For the least upper bound ($a = -\infty$), we have

$$\mathcal{U}_{-\infty} = \lim_{a \to -\infty} \mathcal{U}_a = \min\{\mathcal{U}_\mathbb{P}, \mathcal{U}_\mathbb{C}\}$$

In this work, we use $\mathcal{U}_{-\infty}$ to normalize $I(\mathbb{P}; \mathbb{C})$ to ensure that the maximal attainable NMI value is 1. This brings us the benefit of comparable NMI scores for different corpus sizes $k$.

### 3.3 Greedy Cyclic Abstract Segmentation

We now introduce our GreedyCAS approach to segment abstracts of scientific papers. GreedyCAS performs a search, where we first explore the best segmentation of one particular abstract that maximizes $\mathrm{NMI}(\mathbb{P}; \mathbb{C})$, then iterate over all abstracts to perform the same maximization.

Algorithm 1 describes the basic segmentation approach **GreedyCAS-base**. Given the input abstract corpus $\mathbb{A}$, the algorithm greedily searches for the segmentation that leads to the maximal $\mathrm{NMI}(\mathbb{P}; \mathbb{C})$. The output is the optimized segmentation $\mathbb{G}^*$.

Algorithm 2 illustrates the advanced approach **GreedyCAS-NN**, where we first split the abstract corpus $\mathbb{A}$ into a series of chunks (denoted as $\mathbb{A}^{\mathrm{chunk}}$) in size of $c$; then, for each seed abstract $A_{j_i}^s$ sampled from the current chunk $\mathbb{A}^{\mathrm{chunk}}$, we perform embedding-based nearest neighbour (NN) search within the chunk to construct the batch (denoted as $\mathbb{A}_s^{\mathrm{batch}}$) comprising the $b$ most semantically relevant abstracts for $A_{j_i}^s$, by computing the cosine similarity using their abstract embeddings:

$$\forall A \in \mathbb{A}^{\mathrm{chunk}},$$
$$\mathrm{sim}(A_{j_i}^s, A) = \frac{e(A_{j_i}^s) \cdot e(A)}{||e(A_{j_i}^s)|| \cdot ||e(A)||}$$
$$\mathbb{A}_s^{\mathrm{batch}} = \left\{ A \in \mathbb{A}^{\mathrm{chunk}} : \mathrm{rank}\left( \mathrm{sim}(A_{j_i}^s, A) \right) \leqslant b \right\}$$

We use a pre-trained Sentence-BERT model[2] (Reimers and Gurevych, 2019) to acquire the ab-

---

[2]We use the *sentence-transformers* encoder (pre-trained model *all-MiniLM-L6-v2* with model size 80 MB), Apache-2.0 License, available at github.com/UKPLab/sentence-transformers

**Algorithm 1:** GreedyCAS-base: unsupervised cyclic abstract segmentation

**Input:** abstract corpus $\mathbb{A} = \{A_i\}_{i \in \mathbb{N}_{1:k}}$
**Output:** optimized segmentation $\mathbb{G} = \{G^{A_i}\}_{i \in \mathbb{N}_{1:k}}$

1  $\mathbb{A}^{\text{res}} \leftarrow \mathbb{A}$;
2  **while** $\mathbb{A}^{res} \neq \varnothing$ **do**
3  $\quad \mathcal{O}_{\mathbb{A}}^* \leftarrow 0$;  $\qquad \qquad \triangleright$ optimization objective.
4  $\quad \mathbb{G}^* \leftarrow \varnothing$;
5  $\quad A_i \leftarrow \text{sample}(\mathbb{A})$;
6  $\quad \mathbb{A}^{\text{res}} \leftarrow \mathbb{A}^{\text{res}} \backslash A_i$;
7  $\quad \{g_j^{A_i}\}_{j=1}^6 \leftarrow \text{configure}(A_i)$;
8  $\quad G^{A_i} \leftarrow \{g_j^{A_i}\}_{j=1}^6$;
9  $\quad$ **foreach** *epoch* **do**
10 $\quad \quad \mathbb{G}^{\mathbb{A}^{\text{res}}} \leftarrow \varnothing$;
11 $\quad \quad$ **foreach** $A_r \in \mathbb{A}^{res}$ **do**
12 $\quad \quad \quad \{g_j^{A_r}\}_{j=1}^6 \leftarrow \text{configure}(A_r)$;
13 $\quad \quad \quad g_j^{A_r} \leftarrow \text{sample}(\{g_j^{A_r}\}_{j=1}^6)$;
14 $\quad \quad \quad G^{A_r} \leftarrow \{g_j^{A_r}\}$;
15 $\quad \quad \quad \mathbb{G}^{\mathbb{A}^{\text{res}}} \leftarrow \mathbb{G}^{\mathbb{A}^{\text{res}}} \cup G^{A_r}$;
16 $\quad \quad$ **end**
17 $\quad \quad$ **foreach** $g_j^{A_i} \in G^{A_i}$ **do**
18 $\quad \quad \quad g_j^{A_i} = (P_j^{A_i}, C_j^{A_i})$;
19 $\quad \quad \quad$ **foreach** $G^{A_r} \in \mathbb{G}^{\mathbb{A}^{res}}$ **do**
20 $\quad \quad \quad \quad g_j^{A_r} \leftarrow G^{A_r}$;
21 $\quad \quad \quad \quad g_j^{A_r} = (P_j^{A_r}, C_j^{A_r})$;
22 $\quad \quad \quad$ **end**
23 $\quad \quad \quad \mathbb{P} = P_j^{A_i} \cup \{P_j^{A_r}\}_{A_r \in \mathbb{A}^{\text{res}}}$;
24 $\quad \quad \quad \mathbb{C} = C_j^{A_i} \cup \{C_j^{A_r}\}_{A_r \in \mathbb{A}^{\text{res}}}$;
25 $\quad \quad \quad \mathcal{O}_{\mathbb{A}} \leftarrow \text{compute-NMI}(\mathbb{P}; \mathbb{C})$;
26 $\quad \quad \quad$ **if** $\mathcal{O}_{\mathbb{A}} > \mathcal{O}_{\mathbb{A}}^*$ **then**
27 $\quad \quad \quad \quad \mathcal{O}_{\mathbb{A}}^* \leftarrow \mathcal{O}_{\mathbb{A}}$;
28 $\quad \quad \quad \quad \mathbb{G}^* \leftarrow \mathbb{G}^{\mathbb{A}^{\text{res}}} \cup G^{A_i}$;
29 $\quad \quad \quad$ **end**
30 $\quad \quad \quad$ **else**
31 $\quad \quad \quad \quad$ continue;
32 $\quad \quad \quad$ **end**
33 $\quad \quad$ **end**
34 $\quad$ **end**
35 **end**
36 **return** $\mathbb{G}^*$;

---

**Algorithm 2:** GreedyCAS-NN: unsupervised cyclic abstract segmentation with nearest neighbor search

**Input:** abstract corpus $\mathbb{A} = \{A_i\}_{i \in \mathbb{N}_{1:k}}$, chunk size $c$, batch size $b$
**Output:** optimized segmentation $\mathbb{G}^* = \{G^{A_i}\}_{i \in \mathbb{N}_{1:k}}$

1  $\mathbb{G}^* \leftarrow \varnothing$;
2  $\{\mathbb{A}_i^{\text{chunk}}\}_{i \in \mathbb{N}_{1:k/c}} \leftarrow \text{truncate}(\mathbb{A}, c)$;
3  $\mathbb{A}^{\text{chunk}} \leftarrow \{\mathbb{A}_i^{\text{chunk}}\}_{i \in \mathbb{N}_{1:k/c}}$;
4  **while** $len(\mathbb{G}^*) \neq k$ **do**
5  $\quad$ **foreach** $\mathbb{A}_i^{chunk} \in \mathbb{A}^{chunk}$ **do**
6  $\quad \quad \mathbb{G}_*^{\text{chunk}} \leftarrow \varnothing$;
7  $\quad \quad \{A_{j_i}^s\}_{j \in \mathbb{N}_{1:c/b}} \leftarrow \text{sample}(\mathbb{A}_i^{\text{chunk}})$;
8  $\quad \quad \mathbb{A}_i^s \leftarrow \{A_{j_i}^s\}_{j \in \mathbb{N}_{1:c/b}}$;  $\qquad \triangleright$ seed abstracts
9  $\quad \quad$ **foreach** $A_{j_i}^s \in \mathbb{A}_i^s$ **do**
10 $\quad \quad \quad \{A_{j_i m}^s\}_{m \in \mathbb{N}_{1:b}} \leftarrow \text{NN-search}(A_{j_i}^s; b)$;
11 $\quad \quad \quad \mathbb{A}_s^{\text{batch}} \leftarrow \{A_{j_i m}^s\}_{m \in \mathbb{N}_{1:b}}$;
12 $\quad \quad \quad \mathbb{G}_*^{\text{batch}} \leftarrow \text{GreedyCAS-base}(\mathbb{A}_s^{\text{batch}})$;
13 $\quad \quad \quad \mathbb{G}_*^{\text{chunk}} \leftarrow \mathbb{G}_*^{\text{chunk}} \cup \mathbb{G}_*^{\text{batch}}$;
14 $\quad \quad$ **end**
15 $\quad \quad \mathbb{G}^* \leftarrow \mathbb{G}^* \cup \mathbb{G}_*^{\text{chunk}}$;
16 $\quad$ **end**
17 **end**
18 **return** $\mathbb{G}^*$;

---

stract embeddings. Finally, the same greedy strategy as described in GreedyCAS-base is applied to find the best segmentation for each abstract of the batch. To fully utilize the power of parallel computing, we use multi-threading[3] to optimize NMI.

## 4   Dataset

Since we calculate NMI scores using lexical co-occurrences of words, we constructed a corpus of related abstracts based on the COVID-19 Open Research Dataset (CORD-19) released by Wang et al. (2020). This dataset is a massive collection of sci-

---

entific papers on SARS-CoV-2 coronavirus-related research published since March 2020. These papers share higher lexical commonality than biomedical papers in general due to the focused research interest in COVID-19.

We worked with abstracts whose sentences have been categorized into BACKGROUND, METHODS, RESULTS, and CONCLUSION discourse categories. We trusted the categories of these structured abstracts from the CORD-19 corpus since scientific papers are peer-reviewed and multi-round revised. We automatically aggregated the dataset **CAS-auto** from 697 structured scientific abstracts whose paper titles contained the keyword *vaccine*. Inspired by Shieh et al. (2019), we took sentences in BACKGROUND, METHODS, RESULTS categories as premises, and sentences in the CONCLUSION category as conclusions.

In addition, we manually constructed a dataset **CAS-human** of 196 non-structured abstracts from CORD-19, using the keyword *antigen* to find target abstracts. We then asked four human annotators to label the conclusion sentences within those abstracts. All human annotators were not instructed about the potential positions of conclusion sentences in scientific abstracts. By doing this, we avoided biasing them. To facilitate the annotation process and reduce the annotators' workload, we

---

[3]We use the Python MultiThreading library https://docs.python.org/3/library/threading.html

used the interactive data labeling platform Doccano[4] (Nakayama et al., 2018) for constructing the CAS-human dataset.

Table 2 shows the overall statistics of our proposed datasets for scientific abstract segmentation. During data preprocessing, we intentionally removed stop words, numbers, and punctuations (except ".", which is essential for the sentence tokenizer[5] we used) in the abstracts. We also lowercased all tokens in both datasets for increased computational efficiency.

| Dataset | # abs. | # con. | # pre. | avg. \|abs.\| |
|---|---|---|---|---|
| CAS-auto | 697 | 1,267 | 4,755 | 8.64 |
| CAS-human | 196 | 263 | 1,220 | 7.57 |

Table 2: Statistics of the two datasets. # abs. denotes the number of abstracts, # con. and # pre. indicate the number of conclusion and premise sentences, respectively. avg. \|abs.\| the average number of sentences of the abstracts.

Figure 3 shows the positions of the conclusion sentences within the abstracts in the CAS-human dataset as labelled by the human annotators. Similarly, as shown in previous works (Fergadis et al., 2021; Achakulvisut et al., 2019), in 95% of our annotated non-structured abstracts, the positions of conclusion sentences were consistent with our prior assumption (they were among the last 3 sentences of the abstract).

# 5 Evaluation

## 5.1 Metrics

To evaluate the segmentation results, we use both set similarity and textual relevance as metrics. We test the performance of our approaches on both automatically (CAS-auto) and manually (CAS-human) aggregated data. To evaluate the segmentation boundaries, we use text segmentation metrics $P_k$ (Beeferman et al., 1999) and WindowDiff (WD, Pevzner and Hearst (2002)). Then, we use ROUGE score (Lin, 2004) to measure the textual relevance between the segmented and ground-truth conclusion sentences. We compute the arithmetic mean of ROUGE-1, ROUGE-2, and ROUGE-Lsum f-

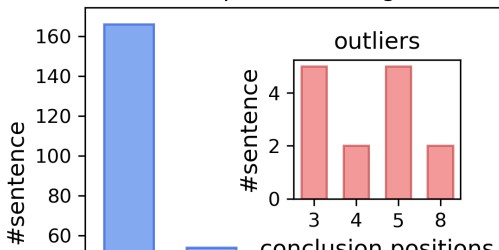

Figure 3: Statistics of the positions of the conclusion sentences within the abstracts in the CAS-human dataset. The minus sign denotes the positions counting from the end of the abstract (-1 denotes the last sentence of the abstract, -2 the second last sentence, and so on).

measures. Finally, we use Jaccard index to measure the similarity between the set of segmented conclusion sentences and the set of ground-truth conclusion sentences. Lower $P_k$ and WD scores indicate better segmentation results, whereas higher ROUGE and Jaccard indexes represent better segmentation results.

## 5.2 Baselines

We present three unsupervised baseline methods for abstract segmentation as baselines. To ensure the comparability of the results, we manually added an additional segmentation boundary at the end of the abstract for any approach that provides only one boundary.

**Random** To test our prior knowledge of the position of conclusions sentences, we set up two random baselines: a) *Random-base*: following the idea initially proposed by Beeferman et al. (1999), we place segmentation boundaries after two randomly selected sentences; b) *Random-plus*: we segment an abstract by randomly selecting one from the six possible segmentations described in chapter 3.1.

**TextTiling[6]** As proposed in Hearst (1997) and serving as the classic text segmentation approach, TextTiling utilizes lexical information to detect topic changes within a given text. In our case, Text-

---

[4]MIT License, available at https://github.com/doccano/doccano

[5]We used the sentence-splitter by Philipp Koehn and Josh Schroeder, GNU Lesser General Public License, available at https://github.com/mediacloud/sentence-splitter

[6]We use the *HarvestText* implementation for the English language, MIT License, available at https://github.com/blmoistawinde/HarvestText

Tiling places a segmentation boundary between sentences.

**SBERT-sim** Inspired by Solbiati et al. (2021), we use the same Sentence-BERT encoder as GreedyCAS-NN to segment abstracts using sentence semantics. Each abstract is split into two segments such that the cosine similarity of their Sentence-BERT embeddings is maximized.

# 6 Results and Discussion

## 6.1 Non-structured Abstracts

First, we test our supervised GreedyCAS approaches against the baselines on the CAS-human dataset of non-structured abstracts. We list the experiment results in Table 3. For the GreedyCAS approaches, we report the empirical performance with the best batch size. The best batch size hints at how many closely related abstracts together can bring benefits to the estimation of word probabilities, and essentially, the final segmentation results. Intuitively, as the batch size grows, the relatedness among the abstracts drops because it is not possible to get a large number of abstracts that study precisely the same research question. For the Random baselines, we report the best segmentation results from 11 random trials.

| CAS-human | $P_k \downarrow$ | WD $\downarrow$ | Jaccard $\uparrow$ | ROUGE $\uparrow$ |
|---|---|---|---|---|
| Random-base | .4293 | .5236 | .0642 | .1925 |
| Random-plus | .2594 | .3678 | .4724 | .6115 |
| TextTiling | .2555 | .3444 | .4971 | .6153 |
| SBERT-sim | .3009 | .4199 | .4491 | .5935 |
| GreedyCAS-base[10] | .1937 | .3089 | .5670 | .6631 |
| GreedyCAS-NN[12] | **.1605** | **.2543** | **.6020** | **.6668** |

Table 3: Segmentation results on the CAS-human dataset. $\downarrow$ indicates that the lower the value is, the better the performance, whereas $\uparrow$ means the opposite. The superscripts of GreedyCAS approaches indicate the best empirical batch size. The best results are statistically significantly better than the closest baseline (Wilcoxon signed-ranked test).

We see that the GreedyCAS-NN achieves the best performance. The results suggest that this approach works well on non-structured abstracts. Thus, NMI is able to capture the conclusion-relevant information at both the abstracts' beginning and end.

## 6.2 Structured Abstracts

Next, we test GreedyCAS approaches against the baselines on the CAS-auto dataset of structured

abstracts. The results are shown in Table 4.

| CAS-auto | $P_k \downarrow$ | WD $\downarrow$ | Jaccard $\uparrow$ | ROUGE $\uparrow$ |
|---|---|---|---|---|
| Random-base | .3958 | .4090 | .1169 | .2578 |
| Random-plus | .2002 | .2251 | .5171 | .6569 |
| TextTiling | .2742 | .3131 | .4009 | .5271 |
| SBERT-sim | .1930 | **.2101** | **.6013** | **.7274** |
| GreedyCAS-base[8] | .1656 | .2341 | .4878 | .5836 |
| GreedyCAS-NN[12] | **.1652** | .2317 | .4830 | .5717 |

Table 4: Segmentation results on the CAS-auto dataset. The best results are statistically significantly better than the closest approach.

First, we found that compared to the Random-base baseline, Random-plus improves the results across different measures by large margins.

We then observe that the SBERT-sim baseline, which segments abstracts based on the cosine similarity between the premise and conclusion segments, achieves the best performance on three out of four metrics by a large margin. Our best model GreedyCAS-NN only achieves the leading performance measured by $P_k$, while achieving lower performance measured by other metrics. In Table 5 and 6 in appendix B, we show two abstracts that were wrongly segmented by GreedyCAS-NN: the first abstract has one additional sentence from the BACKGROUND category, whereas the second abstract has one additional sentence from the RESULTS category. These sentences were misattributed by GreedyCAS to the conclusion segment because they increase the NMI score; however, due to the complexity of NMI, understanding the exact reasons why NMI increases is non-trivial.

## 6.3 Analysis

In Figure 4, we compute the correlation coefficients between NMI scores and each evaluation metric. We plot the NMI scores and the evaluation metrics w.r.t the batch size (ranging from 2 to 12).

Figure 4 shows that NMI scores are strongly negatively correlated with the text segmentation metrics $P_k$ and WindowDiff, but are strongly positively correlated with the set similarity metric Jaccard index and the lexical metric ROUGE.

In Figure 5, rather than studying the complex scenario where an entire sentence gets re-attributed, we showcase the impact on NMI when just one word is moved from the premise segment to the conclusion segment. We studied whether placing segmentation boundaries between words can provide similar NMI$(\mathbb{P}; \mathbb{C})$ scores compared to placing

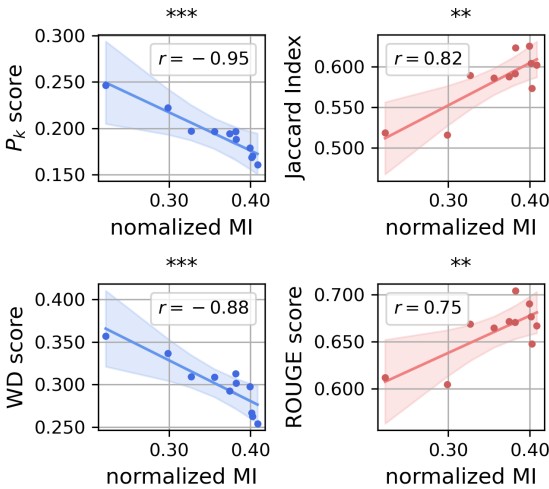

Figure 4: Correlation coefficients between NMI and other metrics. We fit linear regression models to the data points. *** indicates the significance level $p < 10^{-3}$ and ** $p < 10^{-2}$ (Pearson correlation test).

them within sentences. To do this, we randomly pick one abstract from the CAS-human dataset and calculate the changes in NMI($\mathbb{P}; \mathbb{C}$) due to the relocation of the segmentation boundary caused by one word at a time moved to the right.

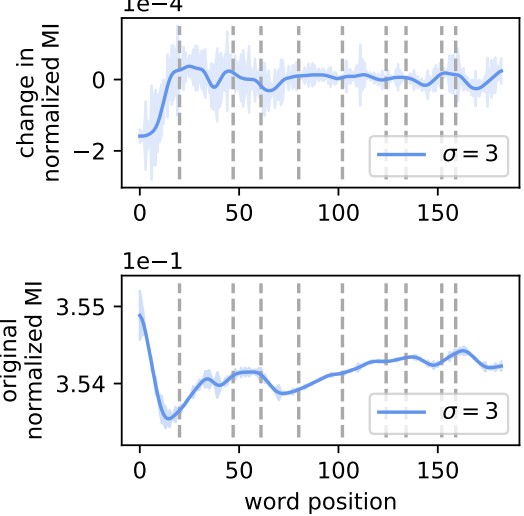

Figure 5: Change in NMI($\mathbb{P}; \mathbb{C}$) when moving one word from the premise to the conclusion segment for a fixed abstract. Dashed lines denote the end positions of sentences in the abstract. We smooth the data with a Gaussian filter with $\sigma = 3$.

We see in Figure 5 that the slope at word positions near sentence boundary is not always steeper than at word positions within the sentences, which indicates that segmenting abstracts by putting seg-

mentation boundaries after complete sentences might not be the optimal choice when optimizing with NMI.

# 7 Conclusion

In this explorative work, we propose an unsupervised approach, GreedyCAS, for automatically segmenting scientific abstracts into conclusions and premises. We introduce the cyclic abstract segmentation pipeline, which can be applied to structured and non-structured abstracts. Our approach leverages the lexical information between words that co-occur in the conclusion and premise segments and finds the best segmentation of a set of abstracts using NMI as an optimization objective. Our empirical results show that NMI is an effective indicator for the segmentation results of scientific abstracts.

# 8 Limitations

In this work, we explored the use of normalized mutual information as an optimization objective for scientific abstract segmentation. The main limitations of our work are listed below:

- The input abstracts of GreedyCAS need to be on similar research topics; otherwise, their shared vocabulary is limited and the word probabilities in the computation of NMI cannot be estimated well.

- GreedyCAS has a high time and space complexity because it involves searching for the best segmentation and enumeration over all possible word pairs at each iteration. As a result, GreedyCAS takes a long time to execute and using larger batch sizes is a challenge with limited computational resources.

- We did not try to assess the reliability of word-pair probability estimation. Presumably, the more abstracts are considered, the better the probability estimates of frequent word pairs, but the more infrequent outlier pairs creep in with biased probability estimates. Thus, it seems not obvious that more abstracts will necessarily yield better probability and mutual information estimates.

In the future, we will analytically study how to increase the efficiency and the applicability of GreedyCAS by considering other ways of estimating the word probabilities.

## 9 Acknowledgements

We acknowledge the support from Swiss National Science Foundation NCCR Evolving Language, Agreement No.51NF40_180888. We also thank the anonymous reviewers for their constructive comments and feedback.

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

# A Exhaustive Search

We tested GreedyCAS on five well-structured abstracts (in total 7,776 configurations) from the CAS-auto dataset, where we use different $a$-orders to normalize the MI. To do this, we brute-forcely calculated NMI values for all configurations. Figure 6 shows the progressive change of the running maximum NMI in the exhaustive search, where the global maximum was reached at around 5,100.th configuration. For all three cases, GreedyCAS managed to reach the same global maximum. However, due to the exponential increase of exhaustive search cost w.r.t number of abstracts, testing GreedyCAS over a larger amount of abstracts was difficult.

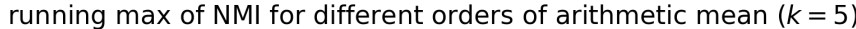

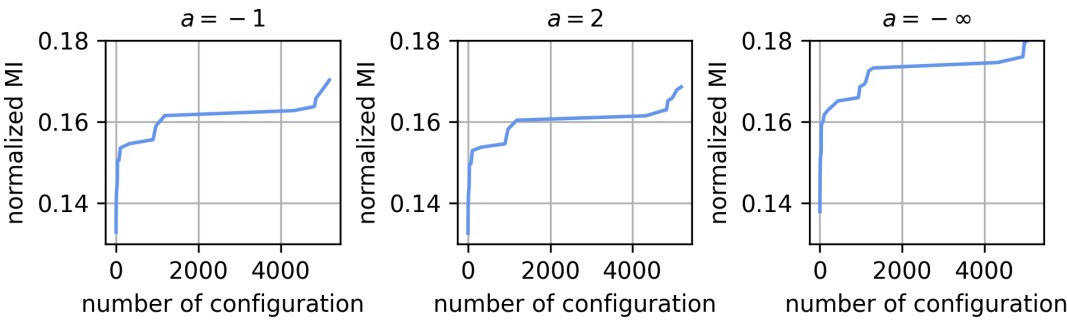

Figure 6: Progressive development of NMI on enumerating 7,776 configurations constructed from $k = 5$ abstracts.

We further tested GreedyCAS on five abstracts with different numbers of trials. Each trial involved different random segmentation initializations. Figure 7 shows the distributions of the number of iterations under different numbers of trials that GreedyCAS required to achieve maximal NMI. We see that GreedyCAS is able to find the maximal NMI within 60 iterations during all trials.

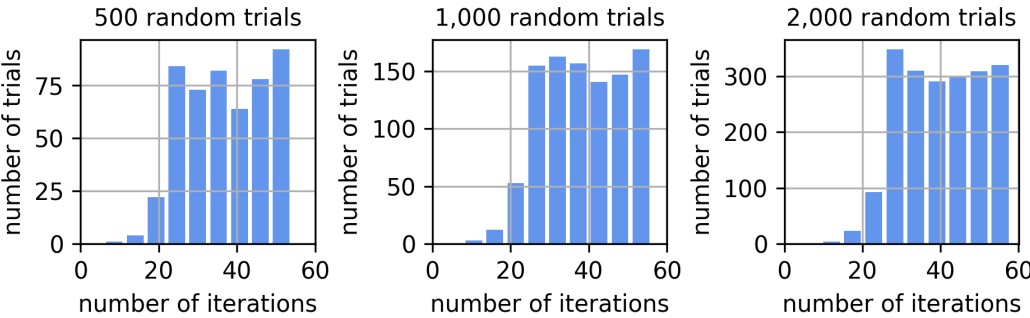

Figure 7: Distribution of number of iterations GreedyCAS performed on five abstracts when reaching the maximal NMI, with different number of random trials. Every trial begins with random initial segmentations of the five abstracts.

## B  Dataset Example

In the following tables, sentences in the premise segment are highlighted in blue, whereas sentences in the conclusion segment are highlighted in red.

---

**Title:** Additional evidence on the efficacy of different Akirin vaccines assessed on *Anopheles arabiensis* (Diptera: *Culicidae*) (Letinić et al., 2021)

**BACKGROUND**  Anopheles arabiensis is an opportunistic malaria vector that rests and feeds outdoors, circumventing current indoor vector control methods. Furthermore, this vector will readily feed on both animals and humans. Targeting this vector while feeding on animals can provide an additional intervention for the current vector control activities. Previous results have displayed the efficacy of using Subolesin/Akirin ortholog vaccines for the control of multiple ectoparasite infestations. This made Akirin a potential antigen for vaccine development against An. arabiensis.

**METHODS**  The efficacy of three antigens, namely recombinant Akirin from An. arabiensis, recombinant Akirin from Aedes albopictus, and recombinant Q38 (Akirin/Subolesin chimera) were evaluated as novel interventions for An. arabiensis vector control. Immunisation trials were conducted based on the concept that mosquitoes feeding on vaccinated balb/c mice would ingest antibodies specific to the target antigen. The antibodies would interact with the target antigen in the arthropod vector, subsequently disrupting its function.

**RESULTS**  All three antigens successfully reduced An. arabiensis and reproductive capacities, with a vaccine efficacy of 68-73%.

**CONCLUSIONS**  These results were the first to show that hosts vaccinated with recombinant Akirin vaccines could develop a protective response against this outdoor malaria transmission vector, thus providing a step towards the development of a novel intervention for An. arabiensis vector control.

---

Table 5: Example abstract in CAS-auto segmented by GreedyCAS-NN, where the first sentence of the BACKGROUND category is attributed to the conclusion segment. Best view in color printing.

---

**Title:** Interest in COVID-19 vaccine trials participation among young adults in China: Willingness, reasons for hesitancy, and demographic and psychosocial determinants (Sun et al., 2021)

**BACKGROUND**  With the demand for rapid COVID-19 vaccine development and evaluation, this paper aimed to describe the prevalence and correlates of willingness to participate in COVID-19 vaccine trials among university students in China.

**METHODS**  A cross-sectional survey with 1,912 Chinese university students was conducted during March and April 2020. Bivariate and multivariate analyses were performed to identify variables associated with willingness to participate.

**RESULTS**  The majority of participants (64.01%) indicated willingness to participate in COVID-19 vaccine trials. Hesitancy over signing informed consent documents, concerns over time necessary for participating in a medical study, and perceived COVID-19 societal stigma were identified as deterrents, whereas lower socioeconomic status, female gender, perception of likely COVID-19 infection during the pandemic, and COVID-19 prosocial behaviors were facilitative factors. Further, public health mistrust and hesitancy over signing informed consent documents had a significant interactive effect on vaccine trial willingness.

**CONCLUSIONS**  High standards of ethical and scientific practice are needed in COVID-19 vaccine research, including providing potential participants full and accurate information and ensuring participation free of coercion, socioeconomic inequality, and stigma. Attending to the needs of marginalized groups and addressing psychosocial factors including stigma and public health mistrust may also be important to COVID-19 vaccine development and future uptake.

---

Table 6: Example abstract in CAS-auto segmented by GreedyCAS-NN, where the last sentence of the RESULTS category is attributed to the conclusion segment. Best view in color printing.

Table 7 shows an example abstract that contains a word pair, whose contribution to the overall NMI

score is the greatest.

| |
|---|
| Title: Hepatitis B surface antigen assembles in a post-ER, pre-Golgi compartment (Huovila et al., 1992) |
| Expression of hepatitis B surface antigen (**HBsAg**), the major envelope protein of the virus, in the absence of other viral proteins leads to its secretion as oligomers in the form of disk-like or tubular lipoprotein particles. The observation that these lipoprotein particles are heavily disulphide crosslinked is paradoxical since **HBsAg** assembly is classically believed to occur in the ER, and hence in the presence of high levels of protein disulphide isomerase (PDI) which should resolve these higher intermolecular crosslinks. Indeed, incubation of mature, highly disulphide crosslinked **HBsAg** with recombinant PDI causes the disassembly of HBsAg to dimers. We have used antibodies against resident ER proteins in double immunofluorescence studies to study the stages of the conversion of the **HBsAg** from individual protein subunits to the secreted, crosslinked, oligomer. We show that **HBsAg** is rapidly sorted to a post-ER, pre-Golgi compartment which excludes PDI and other major soluble resident ER proteins although it overlaps with the distribution of rab2, an established marker of an intermediate compartment. Kinetic studies showed that disulphide-linked **HBsAg** dimers began to form during a short (2 min) pulse, increased in concentration to peak at 60 min, and then decreased as the dimers were crosslinked to form higher oligomers. These higher oligomers are the latest identifiable intracellular form of **HBsAg** before its secretion (t 1/2 = 2 h). Brefeldin A treatment does not alter the localization of **HBsAg** in this PDI excluding compartment, however, it blocks the formation of new oligomers causing the accumulation of dimeric **HBsAg**. Hence this oligomerization must occur in a pre-Golgi **compartment**. These data support a model in which rapid dimer formation, catalyzed by PDI, occurs in the ER, and is followed by transport of dimers to a pre-Golgi **compartment** where the absence of PDI and a different lumenal environment allow the assembly process to be completed. |
| $(w_p, w_c)$ pair that contributes the most to the overall NMI$(\mathbb{P}; \mathbb{C})$ score: (**HBsAg**, **compartment**) |

Table 7: Example abstract in CAS-human segmented by GreedyCAS-NN. The word pair that contributes the most to NMI$(\mathbb{P}; \mathbb{C})$ is in bold. Best view in color printing.

## C Impact on NMI When Moving One Word

Let $w_p \in \mathbb{P}$ be any premise word, $w_c \in \mathbb{C}$ be any conclusion word, $A_i$ a fixed abstract, and $g_j^{A_i}$ one possible segmentation (we use the index $j$ to represent the segmentation). By moving one arbitrary word $w$ from the premise segment $P_j^{A_i}$ to the conclusion segment $C_j^{A_i}$, $I(\mathbb{P}; \mathbb{C})$ changes. We investigate the major terms in the equation of mutual information. We aim to find those word pairs that predominantly contribute to $I(\mathbb{P}; \mathbb{C})$ so that the computation can be simplified, which essentially will allow the algorithm to run on a larger batch size.

First, we examine what happens to the marginal probabilities $p(w_p)$ and $p(w_c)$:

$$p(w_p) = \frac{c(w_p, \mathbb{P}) - \mathbb{I}[w_p = w]}{\sum_{w'_p} c(w'_p, \mathbb{P}) - 1}$$

$$p(w_c) = \frac{c(w_c, \mathbb{C}) + \mathbb{I}[w_c = w]}{\sum_{w'_c} c(w'_c, \mathbb{C}) + 1}$$

Here $\mathbb{I}$ denotes an indicator function and $c$ a counter function. The indicator function takes the value 1 if the condition in the bracket is fulfilled, otherwise, it takes the value 0. For the marginal probabilities, we have the following cases:

- If $w_p = w$, then $p(w_p)$ decreases; if $w_p \neq w$, then $p(w_p)$ increases.

- If $w_c = w$, then $p(w_c)$ increases; if $w_c \neq w$, then $p(w_c)$ decreases.

Then, we examine what happens to $p(w_p; w_c)$, the main term in computation of $I(\mathbb{P}; \mathbb{C})$

$$p(w_p; w_c) = \frac{\sum_{j \neq i} c(w_p, P^{A_j}) c(w_c, C^{A_j}) + \left( c(w_p, P^{A_i}) - \mathbb{I}[w_p = w] \right) \left( c(w_c, C^{A_i}) + \mathbb{I}[w_c = w] \right)}{\sum_{(w'_p, w'_c)} \left( c(w'_p, \mathbb{P}) - \mathbb{I}[w'_p = w] \right) \left( c(w'_c, \mathbb{C}) + \mathbb{I}[w'_c = w] \right)}$$

$$= \frac{\overbrace{\sum_{j \neq i} c(w_p, P^{A_j}) c(w_c, C^{A_j})}^{\text{constant}} + \overbrace{\left( c(w_p, P^{A_i}) - \mathbb{I}[w_p = w] \right)}^{\alpha} \overbrace{\left( c(w_c, C^{A_i}) + \mathbb{I}[w_c = w] \right)}^{\beta}}{\underbrace{\sum_{(w'_p, w'_c)} c(w'_p, \mathbb{P}) c(w'_c, \mathbb{C})}_{\text{constant}} + \underbrace{\sum_{(w'_p, w'_c)} \left\{ c(w'_p, \mathbb{P}) \mathbb{I}[w'_c = w] - c(w'_c, \mathbb{C}) \mathbb{I}[w'_p = w] - \mathbb{I}[w'_p = w] \mathbb{I}[w'_c = w] \right\}}_{\gamma}}$$

$$= \frac{a + \alpha \beta}{b + \gamma},$$

here $a$ and $b$ are the constant terms within the fraction, since moving $w$ in $A_i$ will not affect other abstracts. For the joint probability, we have the following cases:

- If $w_p \neq w$ and $w_c \neq w$, $p(w_p; w_c)$ remains unchanged.

- If $w_p = w$ and $w_c \neq w$, then

$$p(w_p; w_c) = \frac{a + (\alpha - 1)\beta}{b - c(w_c, \mathbb{C})}.$$

- If $w_p \neq w$ and $w_c = w$, then

$$p(w_p; w_c) = \frac{a + \alpha(\beta + 1)}{b + c(w_p, \mathbb{P})}.$$

- If $w_p = w$ and $w_c = w$, then

$$p(w_p; w_c) = \frac{a + (\alpha - 1)(\beta + 1)}{b + c(w_p, \mathbb{P}) - c(w_c, \mathbb{C}) - 1}.$$

Till this point, we found it very difficult to predict how $p(w_p; w_c)$ will change when moving $w$. The reasons are:

- For the case of $w_p = w$ and $w_c = w$, we cannot tell a priori whether $c(w, \mathbb{P}) - c(w, \mathbb{C})$ is positive, i.e. whether $w$ appears more frequently within premise segments or conclusion segments. This leads to the uncertainty of determining the change in the sign of mutual information.

- The normalizing factor of NMI, which is essentially the minimum between $H(\mathbb{P})$ and $H(\mathbb{C})$, cannot be determined after moving $w$.

Also, because for any research domain, it is nearly impossible to get a large number of papers that study exactly the same research question (e.g., it's not possible to get thousands of papers that study the effectiveness of COVID-19 vaccines, due to limited number of clinical trials that have been done so far), therefore, further increasing the batch size is not feasible.

Due to the above reasons, we only computationally studied how NMI would change when moving one word from the premise segment to the conclusion segment (see Figure 5).