# OpenReview forum: "GreedyCAS: Unsupervised Scientific Abstract Segmentation with Normalized Mutual Information"
_EMNLP/2023/Conference — EMNLP 2023 Main_

### Official Review · Reviewer_Wkik · 2023-08-04

**Soundness:** 4

**Excitement:**

4: Strong: This paper deepens the understanding of some phenomenon or lowers the barriers to an existing research direction.

**Paper Topic And Main Contributions:**

This paper addresses the challenging task of automatic segmentation of conclusions from scientific abstracts, specifically focusing on non-structured abstracts. In non-structured abstracts, the concluding information is not explicitly marked.

The main contribution of this paper is the introduction of an unsupervised approach called "GreedyCAS" for abstract segmentation. The approach is based on Normalized Mutual Information (NMI). The paper makes some acceptable assumptions: (1) consider each abstract as a recurrent cycle of sentences and place two segmentation boundaries and (2) it assumes that conclusions are strongly semantically linked with preceding premises.



**Questions For The Authors:**

A: In section 6.3, you mention "the semantic metric ROUGE". However, ROUGE is not explicitly semantic. Have you explored other measures that directly capture semantic similarity?

B: Your proposed approach seems promising for abstract segmentation in scientific papers. Have you conducted any experiments or explored the adaptability of your method to other domains beyond scientific literature?

**Reasons To Accept:**

1. The paper introduces a novel approach. This provides a fresh perspective on the problem and offers a potential solution that is different from existing methods.
2. The proposed approach is well-grounded in theory by utilizing NMI as a measure of semantic similarity between sentences. This theoretical foundation adds credibility to the method and makes it easier for researchers in the NLP community to understand and build upon the proposed technique.
3. The authors have tested their approach using two variants, one based on nearest neighbor (NN) and the other standard (base). Additionally, the method has been extensively evaluated against baselines.

**Reasons To Reject:**

No major issues in the paper.

**Reproducibility:**

4: Could mostly reproduce the results, but there may be some variation because of sample variance or minor variations in their interpretation of the protocol or method.

**Reviewer Confidence:**

3: Pretty sure, but there's a chance I missed something. Although I have a good feel for this area in general, I did not carefully check the paper's details, e.g., the math, experimental design, or novelty.

---

> ### Author Rebuttal · Authors · 2023-08-29
>
> Dear reviewer Wkik,
>
> We sincerely thank you for the valuable comments. Please kindly see our response to your questions as follows:
>
> **Q:  ROUGE is not explicitly semantic, have you explored other measures that directly capture semantic similarity?**
>
> A: Thank you for the precise comment. It's accurate that ROUGE doesn't directly gauge the semantic relevance between the original and segmented conclusions. We will revise the text for improved accuracy.
>
> We opted against using semantic-based evaluation metrics because they often depend on the text encoder chosen. For instance, in our experience, Sent2vec (Matteo Pagliardini, Prakhar Gupta, Martin Jaggi, Unsupervised Learning of Sentence Embeddings using Compositional n-Gram Features NAACL 2018) embeddings for semantically different texts tend to nevertheless have high cosine similarity scores (>0.6). Cosine similarity is thus not discriminative enough for evaluating segmentation boundaries.
>
> **Q: Have you conducted any experiments or explored the adaptability of your method to other domains beyond scientific literature?**
>
> A: Thank you for your interest. Our focus in this paper was on scientific text, but GreedyCAS in principle is applicable to domains beyond scientific literature and we will explore its performance in future work.
>
> We hope that we have addressed your concerns and we are happy to discuss further.

---

### Official Review · Reviewer_RqZY · 2023-08-05

**Soundness:** 4

**Excitement:**

3: Ambivalent: It has merits (e.g., it reports state-of-the-art results, the idea is nice), but there are key weaknesses (e.g., it describes incremental work), and it can significantly benefit from another round of revision. However, I won't object to accepting it if my co-reviewers champion it.

**Paper Topic And Main Contributions:**

This work proposed an unsupervised approach that utilizes mutual information to segment unstructured abstracts into premise (sections other than the conclusion subsection) and conclusion. The study treats every abstract as a recurrent cycle of sentences. To optimally determine segmentation boundaries, it greedily optimizes the NMI score between the two segments, and decides the boundary for the segmentation. The method and evaluation were done on CORD-19 datasets.

**Questions For The Authors:**

a. The current method section needs a bit more detailed description for reproducibility. For example, in section 3.1, the authors argued that 6 possible segmentations per abstract exist in current methods, but it reads that there could be more combinations of boundary and non-boundary sentences in an abstract. Could the authors elaborate more on it?

b. Since the conclusion sentences normally occur towards the end of the abstracts - I wonder if separating the abstracts into two parts (first half and second half) first, and applying the current methods in the section half of the abstracts only would make it more efficient?

c. In section 4, could the authors provide more details of the inter-coder agreement checking results between the annotators when labeling the sections?

d. Could the authors elaborate more on the motivation of why segmenting the abstracts into only two subsections instead of a more fine-grained segmentation?

**Reasons To Accept:**

Segmenting unstructured abstracts or scientific papers into subsections is an important task. Developing methods in this task could potentially be beneficial to the community in other related tasks, such as segmenting electronic health records, etc.

The study proposed a new approach that utilized mutual information for abstract segmentation.

**Reasons To Reject:**

1. Current approach segments the abstracts into two parts - premise and conclusion. However, most early efforts have been tried to segment the abstracts into sections with more fine granularity. Comparatively, just identifying the conclusion from the abstracts might not provide enough information that is needed.

2. As discussed by the authors - the proposed methods has a high time and space complexity. Due to the large number of scientific papers, the efficiency of the method might be an issue in real practice.

**Reproducibility:**

3: Could reproduce the results with some difficulty. The settings of parameters are underspecified or subjectively determined; the training/evaluation data are not widely available.

**Reviewer Confidence:**

3: Pretty sure, but there's a chance I missed something. Although I have a good feel for this area in general, I did not carefully check the paper's details, e.g., the math, experimental design, or novelty.

---

> ### Author Rebuttal · Authors · 2023-08-29
>
> Dear reviewer RqZY,
>
> We sincerely thank you for the valuable comments. We kindly ask you to review our answers to your queries provided below:
>
> **Q: Identifying the conclusion from the abstracts might not provide enough information that is needed.**
>
> A: In this study, our focus does not revolve around categorizing premises into distinct discourse segments (Background, Methods, Results, and Conclusion). Rather, our objective is to investigate whether NMI, relying solely on word statistics, possesses the capability to differentiate between conclusions and premises. Early efforts in discourse sentence classification often introduce external language information not inherent to the abstracts. In contrast, our approach relies solely on the estimation of word probabilities derived from the abstract texts.
>
> **Q: Due to the large number of scientific papers, the efficiency of the method might be an issue in real practice.**
>
> A: Thank you for the comment. The high computational cost of GreedyCAS is largely due to the need for reliable word probabilities, which in turn is only possible when many abstracts are processed simultaneously. In a separate ongoing work, we explore whether using pretrained language models, which have already processed large text corpora, have reliable word probability estimations without needing to process as many abstracts.
>
> **Q: There could be more combinations of boundary and non-boundary sentences in an abstract?**
>
> A: To reduce computational cost, we assumed that 1) abstracts contain at most three conclusion sentences and 2) conclusion sentences are not located within the middle of abstracts. These assumptions lead us to six possible combinations of boundary and non-boundary sentences per abstract. However, without these assumptions, the total number of possible combinations per abstract grows exponentially because the two boundary sentences may be anywhere within the abstract.
>
> **Q: Separating the abstracts into two parts (first half and second half) first, and applying the current methods in the section half of the abstracts only would make it more efficient?**
>
> A: Thank you for this comment. Indeed, adopting your suggestion could enhance efficiency by extracting conclusions solely from the second half of the abstracts. We applied GreedyCAS to the full abstract not for efficiency, but to analyze its ability to segment abstracts under a more difficult setting (where there are more possible conclusion boundary positions to choose from).
>
> **Q: Provide more details of the inter-coder agreement checking results between the annotators when labeling the sections. Why segmenting the abstracts into only two subsections instead of a more fine-grained segmentation.**
>
> A: Within the CAS-human dataset, we refrained from requesting annotators to assign a section (Background, Results, Methods) to each sentence of the abstract. This omission stemmed from the fact that section details were not considered into GreedyCAS's input, and section classification is not the goal of this work.
>
> We hope that we have addressed your concerns and we are happy to discuss further.

---

### Official Review · Reviewer_m93t · 2023-08-13

**Soundness:** 4

**Excitement:**

3: Ambivalent: It has merits (e.g., it reports state-of-the-art results, the idea is nice), but there are key weaknesses (e.g., it describes incremental work), and it can significantly benefit from another round of revision. However, I won't object to accepting it if my co-reviewers champion it.

**Paper Topic And Main Contributions:**

This paper proposes an unsupervised approach for scientific abstract segmentation, named Greedy Cyclic Abstract Segmentation (GreedyCAS), which optimizes Normalized Mutual Information (NMI). The approach uses an exhaustive greedy approach to iterate over all abstracts and determine the best segmentation for each. To test how NMI deals with a known text boundary, the end of an abstract, the start and end of each abstract are stitched together to form a cycle, then two segmentation boundaries are selected with constraints based on prior knowledge. Two datasets are created to test the proposed approach, one comprises non-structured abstracts with human-annotated conclusion sentences, and the other contains structured abstracts in which conclusion sentences have been explicitly marked by the authors of the abstract. The results show that GreedyCAS achieves promising segmentation results on the dataset of non-structured abstracts, and there is a strong correlation between NMI and other evaluation metrics.
Main contributions:
1. Propose GreedyCAS, an unsupervised approach for scientific abstract segmentation that optimizes NMI.
2. Show that GreedyCAS achieves promising segmentation results on a dataset of non-structured abstracts.
3. Find a strong correlation between NMI and other evaluation metrics, in support of the effectiveness of the proposed approach.

**Questions For The Authors:**

1. Is this paper the first to solve the abstract segmentation in an unsupervised manner?
2. Why stitching the start and end together?
3. How to calculate NMI in GreedyCAS-NN?
4. Does the abstract segmentation has publicly available datasets? Why constructing the dataset in your own?

**Reasons To Accept:**

1. The proposed GreedyCAS approach optimizes Normalized Mutual Information (NMI) for unsupervised scientific abstract segmentation, offering a new solution to an important NLP task.
2. Experimental results on two datasets demonstrate the effectiveness of the approach, with promising results on non-structured abstracts and strong correlation between NMI and other evaluation metrics.
3. The discussion of limitations and suggested future work demonstrates the author's commitment to improvement, contributing to the overall quality of the paper.

**Reasons To Reject:**

1. The datasets used for testing are relatively small and do not represent a broad range of possible scenarios.
2. The method is heuristic.
3. For  Structured Abstracts, GreedyCAS seems to have no advantage and the paper does not conduct depth discussions.

**Reproducibility:**

4: Could mostly reproduce the results, but there may be some variation because of sample variance or minor variations in their interpretation of the protocol or method.

**Reviewer Confidence:**

3: Pretty sure, but there's a chance I missed something. Although I have a good feel for this area in general, I did not carefully check the paper's details, e.g., the math, experimental design, or novelty.

---

> ### Author Rebuttal · Authors · 2023-08-29
>
> Dear reviewer m93t,
>
> We sincerely thank you for the valuable comments. We kindly ask you to review our answers to your queries provided below:
>
> **Q: Datasets used are relatively small and do not represent a broad range of possible scenarios.**
>
> A: By definition, NMI requires word probabilities to be well-estimated. To make sure that the word probabilities are reliable, we use abstracts from papers in the CORD19 dataset, since they all focus on covid related research topics, and close-related papers have higher cooccurrences of premise and conclusion words. Nevertheless, to further narrow down the research topics, we fetch structured abstracts whose paper titles contain the keyword *vaccine*, and non-structured abstracts, the keyword *antigen*. This results in having nearly 700 structured abstracts and 200 non-structured abstracts, which is the best we can do with the CORD19 dataset.
>
> **Q: The method is heuristic.**
>
> A: We aim to investigate the suitability of NMI as an optimization objective for segmenting abstracts into premises and conclusions. The assumptions about conclusion sentence location and number rely on heuristics, which are necessary to prevent excessive computational costs in GreedyCAS. Additionally, Figure 3 illustrates that these heuristics closely correspond to human annotations on the location of conclusion sentences, indicating their validity.
>
> **Q: For Structured Abstracts, GreedyCAS seems to have no advantage and the paper does not conduct depth discussions.**
>
> A: Thank you for the comment. When writing structured abstracts, we hypothesize that authors take text directly from the corresponding sections in the full text instead of writing novel text. This may result in the sentences of structured abstracts being less coherent with one another (in the sense of having fewer words in common), which in turn may affect the performance of GreedyCAS. We will edit our manuscript to include this discussion.
>
> **Q: Is this paper the first to solve the abstract segmentation in an unsupervised manner?**
>
> A: To the best of our knowledge, it is. However, we are delighted to know and cite any related works, if you have any suggestions.
>
> **Q: Why stitch the start and end together?**
>
> A: Conclusion sentences are typically located at the end of the abstracts, so segmenting original abstracts into premises and conclusions is often a matter of finding just one conclusion boundary sentence. By stitching the start and end of each abstract together, we essentially evaluate GreedyCAS on the harder task of having to identify both conclusion boundary sentences.
>
> **Q: How to calculate NMI in GreedyCAS-NN**
>
> A: The computation of NMI within GreedyCAS-NN mirrors that of GreedyCAS-base. The distinction, however, lies in the approach. In GreedyCAS-NN, during batch creation, a pre-trained sentence encoder SBERT is employed to retrieve abstracts that carry the greatest semantic relevance to the seed abstract. The rationale here is rooted in the belief that semantically similar abstracts possess a higher degree of shared vocabulary. This, in turn, plays a pivotal role in enhancing the precision of word probability estimation during the NMI calculation process.
>
> **Q: Does the abstract segmentation has publicly available datasets? Why constructing the dataset in your own?**
>
> A: To the best of our knowledge, no existing publicly accessible datasets align precisely with our objectives. To ensure accurate word probability estimation, it's essential to source abstracts that pertain closely to our research themes. Consequently, we made the determination to compile these abstracts on our own.
>
> We hope that we have addressed your concerns and we are happy to discuss further.

---

### Meta-Review · Area_Chair_XPNP · 2023-09-20

**Recommendation:** 5

**Metareview:**

The paper proposes an unsupervised approach for scientific abstract segmentation, and all the reviewers agree its soundness and give comprehensive reviews. The authors also response to the questions in detail. In addition, reviewers also have some concerns, e.g., efficiency. I strongly suggest that the authors make precise revisions based on the comments of the reviewers.

---

### Decision · Program_Chairs · 2023-10-07

**Decision:**

Accept-Main

**Comment:**

The paper proposes an unsupervised approach for scientific abstract segmentation, and all the reviewers agree its soundness and give comprehensive reviews. The authors also response to the questions in detail. In addition, reviewers also have some concerns, e.g., efficiency. I strongly suggest that the authors make precise revisions based on the comments of the reviewers.